# What Is the “Hydrogen Bond”? A QFT-QED Perspective

**DOI:** 10.3390/ijms25073846

**Published:** 2024-03-29

**Authors:** Paolo Renati, Pierre Madl

**Affiliations:** 1World Water Community, NL-3029 Rotterdam, The Netherlands; 2Prototyping Unit, Edge-Institute, ER-System Mechatronics, A-5440 Golling, Austria; 3Department of Biosciences & Medical Biology, University of Salzburg, A-5020 Salzburg, Austria

**Keywords:** quantum field theory, phase, coherence, water, symmetry-breaking, dynamical order, resonance, non-thermal effects, hydrogen bond

## Abstract

In this paper we would like to highlight the problems of conceiving the “Hydrogen Bond” (HB) as a real short-range, directional, electrostatic, attractive interaction and to reframe its nature through the non-approximated view of condensed matter offered by a Quantum Electro-Dynamic (QED) perspective. We focus our attention on water, as the paramount case to show the effectiveness of this 40-year-old theoretical background, which represents water as a two-fluid system (where one of the two phases is coherent). The HB turns out to be the result of the electromagnetic field gradient in the coherent phase of water, whose vacuum level is lower than in the non-coherent (gas-like) fraction. In this way, the HB can be properly considered, i.e., no longer as a “dipolar force” between molecules, but as the phenomenological effect of their collective thermodynamic tendency to occupy a lower ground state, compatible with temperature and pressure. This perspective allows to explain many “anomalous” behaviours of water and to understand why the calculated energy associated with the HB should change when considering two molecules (water-dimer), or the liquid state, or the different types of ice. The appearance of a condensed, liquid, phase at room temperature is indeed the consequence of the boson condensation as described in the context of spontaneous symmetry breaking (SSB). For a more realistic and authentic description of water, condensed matter and living systems, the transition from a still semi-classical Quantum Mechanical (QM) view in the first quantization to a Quantum Field Theory (QFT) view embedded in the second quantization is advocated.

## 1. Introduction

The existence of “forces” deemed responsible for cohesion between molecules such as H_2_O, HF, NH_3_ or many other compounds such as carboxylic acids, amino acids, etc., was first conceived and realized in the early decades of the last century, when it was not clear why the melting points and latent heats of water, ammonia, or hydrogen fluoride were higher than those of other hydrides of heavier elements (such as H_2_S, HCl, PH_3_, SiH_4_, etc.). Studies at that time suggested that the higher the molecular weight, the higher the boiling or melting point and the latent heat of vaporization should be [1,2,3]. Furthermore, after the discovery of X-ray diffraction, it became clear that, in ice, water molecules are strongly arranged in a tetrahedral crystalline structure, characterized by large hexagonal channels in which various gases can be trapped [4].

When Huggins first introduced the HB in 1921 [5], it was a concept that fits into the prevailing paradigm. However, it was not foreseeable that this notion would become a central pillar of biochemistry, ignoring the fact that contemporary physics was about to offer a more elegant and better fitting concept. Carl Linus Pauling eventually gave the HB some “chemical respectability”, moving from an earlier static conception [6,7] to a more dynamic picture, based on resonance hybrids of molecules where the delocalization of electrons provides thermodynamical stability to the HBs (or H-bridges) [8]. Similarly, the seminal work of Bernal and Fowler [9] promoted that the liquid nature of water is the result of the HB-ing network and, therefore, proposed a large-scale adaptation of the HB model in physical and chemical applications. Eventually, by using electrically neutral molecular formulae for the water molecule, Pauling published a paper claiming that the structure and residual entropy of hexagonal ice is related to the intrinsic asymmetry of the HB itself [10]. It was William H. Zachariasen, in his 1935 study of the structure of liquid methyl alcohol, who first suggested the dipolar nature of the HB [11].

To justify the strong attraction that raises the boiling point of water from −150 °C (in the absence of the postulated HB) to +100 °C (based on the experimental value), most scientists prefer to focus on the average number of electrons around the nuclei. Quantum analysis shows a slight excess of electrons around the oxygen atom (note δ−) combined with a slight deficit around the hydrogen atom (note δ+), all of which is consistent with Zachariasen’s interpretation. However, as we shall see later, this does not solve the problem, since the quantum-calculated polarity of such a charge configuration gives a relative dielectric constant of liquid water that differs substantially from the experimental values. Nevertheless, the early studies were so extensively used to establish the arbitrary concept of the HB, that it became—and still is—widely used in chemistry and especially in the biological sciences. Thus, in 2011, the International Union of Pure and Applied Chemistry (IUPAC) provided an “official definition” of the HB (quoted below), which is accompanied by twelve “emendations” listed therein [12].

Most of the 12 criteria are purely empirical (see E3 to E6 and C1 to C6 in the reference cited above, or in the Appendix A). These tell us absolutely nothing about the causes behind the observed closeness/attraction between atoms X and Y, as they are purely technical criteria based on a variety of independent observations. Moreover, the remaining statements provided by the IUPAC (El and E2) claim (i) the existence of “forces” based on a purely electrostatic nature of the interaction between electric charges and (ii) the “covalence” of such bonds, which is refuted by refined experimental evidence, quoted below. The fact that it takes such a long time to define a single concept clearly underlines its weak foundations. This model, as it stands, is only able to explain how the formation of alpha-helices or beta-sheets in proteins is stabilized by the HB, as originally proclaimed by Pauling [13]. These inconsistencies led Marc Henry to state that the HB remains by definition “a guest without a face” [14]. The same author, in recent years, has written important contributions [14,15,16] about the problems related to the description of the condensed states of water through the corpuscular view of QM (first quantization), which is based on the idea of the HB conceived as a local, directional, and electrostatic force. Here, for the first time to our knowledge, we gather in a coherent and handy review all the key aspects that haunt the QM perspective and an effective exposition of how QFT-QED reframes the picture, providing also visual sketches of the energetics related to the coherent fraction in the QED two-fluid portrait of liquid water.

## 2. Theoretical Background and Comments on Experimental Data

### 2.1. Recap of the Main Problems within the Corpuscular QM View (First Quantization)

In fact, the whole picture on which Pauling, the IUPAC, and the most chemists base their positions is that of a corpuscular description of matter (both classical and quantum mechanical, QM), which is incapable of explaining the tricky features of condensed systems like liquid water. The problem is that, according to Maxwell’s equations (Langevin’s theory), liquid water should have a relative dielectric constant given by:(1)$$\varepsilon_{r}=1+\frac{N_{A}\times \rho \times p_{0}^{2}}{3k_{B}T\times \varepsilon_{0}}=1+\frac{18500\times [\rho /g\times {cm}^{-3}]\times {\left[p_{0}/D\right]}^{2}}{[T/K]\times [M/g\times {mol}^{-1}]}$$

Having *N_A_*, the Avogadro number, *p*_0_ = 1.85498 D (D: Debye, such as 1 D = 3.336 · 10^−30^ C·m), *ρ* = 1 g·cm^−3^ and *M* = 18 g·mol^−1^, a first problem relates to the fact that Equation (1) gives a value for *ε_r_* (T = 300 K) ≈ 13 instead of the experimentally obtained *ε_r_* = 80 for bulk water. Furthermore, this approach cannot explain why the solubility of water for most ionic salts increases with temperature while the dielectric permittivity decreases [17]. To obtain the electrical stiffness of water molecules (with polarizability volume α′ = 1.47 Å^3^, ionization potential, or ionization threshold, I_th_ = 12.6 eV, molecular radius R = v2.4 Å), Maxwell’s equations suggest dipolar interaction energies [18] equal to:0.05 eV for direct dipole-dipole Keesom interactions (Equation (2))
(2)$$U_{dd}=-\frac{2p_{0}^{4}}{3{\left(4\pi \varepsilon_{0}\right)}^{2}\cdot k_{B}T\cdot R^{6}}\Rightarrow U_{dd}=-3013.81\frac{{{(p}_{0}/D)}^{4}}{(T/K)\cdot {(R/Å)}^{6}}\approx 0.05\;eV$$

0.03 eV for Debye interactions between permanent and induced dipoles (Equation (3))


(3)
$$U_{ind}=-\frac{2\alpha \cdot p_{0}^{2}}{{\left(4\pi \varepsilon_{0}\right)}^{2}\cdot R^{6}}\Rightarrow U_{ind}=-1.2483\frac{{(\alpha /{Å}^{3})\cdot {(p}_{0}/D)}^{2}}{{(R/Å)}^{6}}\approx 0.03\;eV$$


0.12 eV for dispersive London interactions between two induced dipoles (Equation (4))


(4)
$$U_{displ}=-\frac{3\alpha^{2}\cdot I}{{4(4\pi \varepsilon_{0})}^{2}\cdot R^{6}}\Rightarrow U_{displ}=-0.75\frac{{\left(\alpha /{Å}^{3}\right)}^{2}}{{(R/Å)}^{6}}\cdot (I/eV)\approx 0.12\;eV$$


These calculations assume an oxygen–oxygen distance of 3.65 Å, corresponding to the O-H covalent bond length (0.95 Å), plus the sum of the van der Waals radii of hydrogen (1.2 Å) and oxygen (1.5 Å). Such values cannot explain the unusually high boiling point of liquid water, or the HB energy of about 0.2 eV. Even summing these classical-theory values does not help, since the HB energy depends on the molecular environment of the water molecules. For example, the HB energy is reported to be 0.15 eV in the “water dimer”, 0.24 eV in liquid water and 0.29 eV in hexagonal ice [19] and its apparent covalency casts doubt on its electrostatic nature [20]. The problem is that it is impossible to measure each classical/QM interaction (Keesom, Debye and London) separately, they can only be evaluated by using a theoretical model and these interactions would remain exactly the same whatever the aggregation state of the water molecules is. Thus, from an energetic point of view, HBs behave quite differently from van der Waals forces (Keesom, Debye, London) [15].

Especially for liquid water, the agreement of theoretical models—based on the QM picture of matter—with the experimental results and the known water anomalies is far from satisfactory. In this context we quote Henry [14]:


*“The claim that chemistry has been completely explained in terms of quantum theory is now received wisdom among physicists and chemists. Yet quantum physics is able neither to predict nor explain the strong association of water molecules in liquid or ice. Quantum chemistry algorithms either exclude hydrogen bonded (H-bonded) systems, or treat them by modelling a water molecule as an asymmetric tetrahedron having two positive and two negative electrical charges at its vertices. Recent calculations of the potential energy surface of the simple water dimer {H_2_O}_2_ yield 30,000 ab initio energies at the coupled clustering techniques (CCT) level [21]. But free OH-stretches [deviate from] experimental values by 30–40 cm^−1^ and their dissociation energy 1.1 kJ·mol^−1^ [are likewise] below benchmark experimental values. To obtain satisfactory agreement with experiment, it is necessary to replace ab initio potentials with spectroscopically accurate measurements. This is hardly a ringing endorsement of the underlying theory”*
(despite Dirac’s 1929 claims [22]).

In fact, molecular orbital (MO) theory (basically an application of a QM-first quantization theory to the molecular orbital approximations) studies molecular bonding by approximating the positions of bonded electrons by a linear combination of their atomic orbitals (LCAO). This is achieved, for example, by applying the Hartree-Fock model to Schrödinger’s equation [18]. But within the LCAO picture there are big problems regarding the topology and overlapping of the orbitals of the water molecule. If we consider the basic C_2v_-symmetry of H_2_O (according to the Schönflies classification [23]), water has four irreducible representations called a_1_, a_2_, b_1_ and b_2_, where “a” (“b”) indicates symmetric (anti-symmetric) representation with respect to a rotation around the main symmetry axis, in this case the *z*-axis, the same one along which the *p_z_* oxygen orbital is oriented. The subscripts “1” and “2” indicate, respectively, symmetric and antisymmetric representations with respect to the rotation about a C_2_ axis, perpendicular to the main symmetry axis, or about a plane σ_v_, if C_2_ is missing.

In a single water molecule, we have ten electrons (their occupation number for each orbital is given by the superscript out of the brackets in the following expression) that must be distributed among five energy levels according to the electronic configuration: (1a_1_)^2^(2a_1_)^2^(1b_2_)^2^(3a_1_)^2^(1b_1_)^2^(4a_1_)^0^(2b_2_)^0^ (see Figure 1).

Accordingly, the partial covalence involving the Highest Occupied Molecular Orbital (HOMO), which has b_1_-symmetry (non-symmetric with respect to the *z* axis—the same along which is oriented the *p_z_* oxygen orbital) and the Lowest Unoccupied Molecular Orbital (LUMO), which displays a_1_-symmetry (symmetric with respect to the *z* axis)—cannot be established. Thus, no HOMO/LUMO interaction overlap can possibly occur.

Even the argument that during hydrogen bonding, the symmetry is lowered, thus leading to a possible non-zero overlap, is still unsatisfactory, because prior to the HB both partners show their full C_2v_-symmetry with zero overlap, whereas we know from experiments that the final symmetry of water dimers, or more numerous aggregates, is C_s_ (reflection with respect to a *σ*-plane). So, at what distance would the symmetry change from C_2v_ to C_s_? The assumption that C_s_-symmetry would be maintained at any distance is useless because the HOMO level would still represent one symmetry, and the LUMO another: the overlapping integral would be again zero. One might think that overlaps could occur through other molecular orbitals, describing the covalent O-H as *σ*-bonds, leaving two outer non-equivalent “lone-pairs” (3a_1_, 1b_1_) available to form HBs with other water molecules, but at both 2.75 Å (the distance reached by 3a_1_) and 2.98 Å (the distance reached by 1b_1_), the overlap between the acceptor oxygen and the HB-ing proton is negative, because the 3a_1_ (HOMO−1) and the 1b_1_ (HOMO) have two very different topologies and energies, indicating a net anti-bonding covalent interaction in the quantum sense [19]. Note: (HOMO−1, HOMO−2, HOMO−3, …, HOMO−N or LUMO+1, LUMO+2, LUMO+3, …, LUMO+N) denote electronic levels, among the several molecular orbitals, placed at the N^th^ level below (−) the HOMO or at the N^th^ level above (+) the LUMO).

Furthermore, X-ray emission spectroscopy (XES) reveals that in a water molecule the 1b_1_ HOMO-level is not affected by the HB [26], whereas a strong perturbation of the 3a_1_ (HOMO-1) level is observed. This is evidence for a rather unconventional (within the LCAO QM picture) HOMO-1/LUMO interaction (see Figure 2). In addition, Compton scattering experiments which revealed a strong anisotropy of the momentum density of the valence electrons in hexagonal ice (I_h_-type), are evidence of a neat anti-bonding, repulsive interaction between neighbouring water molecules despite the multicentred character of the QM wave functions [27]. Finally, topological analysis of the electronic density showed that it was not possible to distinguish between HBs and pure van der Waals interactions [28].

Given the experimental photoelectron spectrum of the water molecule (Figure 3), its most faithful representation should display three types of orbitals (two σ-bonds, one 2s-type lone pair and one 2p-type lone pair), rather than two types (two σ-bonds and two equivalent lone pairs), as suggested by MO theory [30]. The only way to retrieve a physical picture involving two lone pairs and two σ-bonds oriented approximately towards the vertices of a tetrahedron, is to look at the positions of the largest eigenvalues and corresponding eigenvectors of the Hessian minima in their molecular electrostatic potential [31]. But, again, this means reverting again to a purely electrostatic view of HBs with all the associated problems we are listing here.

The situation is so confusing that the scientific community today is now divided into two opposing camps, one promoting water as a random tetrahedral network with flickering HBs [34], and the other promoting water in terms of a two-state model, one tetrahedral and the other not [26,35]. The first picture contradicts a very different picture obtained when X-rays are absorbed by water, where the water does not appear to be arranged in a local tetrahedral geometry, but rather as entangled chains [35]. The usually adopted two-state model is closer to reality, but the underlying idea on which it is based—still corpuscular QM—implies a physically unmotivated “cut-off energy” at which the two populations of molecules should separate (this point will be elaborated further below). If we imagine liquid water as a flickering network of HBs, the problem lies in the way this interaction is usually conceived and treated in the theoretical models: electrostatically or electrodynamically but still in a perturbative way and only in a first quantization. Indeed, neutron scattering experiments, as well as molecular dynamic simulations, have shown that the average residence-time of a hydrogen atom around a water molecule is, in average, about 2 ps at T = 300 K, and increases to 20 ps at T = 250 K [36]. Note: The problem with X-ray scattering, is that it tends to give a static image of water, whereas it is a dynamic medium. Neutron scattering, on the contrary, reveals the existence of two relaxation times in liquid water [36]. The first, close to 1–2 ps at room temperature, corresponds to the fluctuation of the HB-network following the rotations of the water molecules. This relaxation time follows an Arrhenius law τ_LH_ = τ_0_∙exp(U^#^/k_B_T) with τ_0_ = 0.0485 ps and an activation energy U^#^ = 7.7 kJ/mol. As for the second relaxation time, it varies very strongly with temperature, from 1.25 ps at 20 °C to 22.7 ps at −20 °C. This indicates that there are two fractions, one of which has intrinsic dynamics independent of temperature (the coherent fraction, as it will be discussed detail below). 

Electric charges moving on the picosecond timescale are expected to generate an electromagnetic field with a frequency of at least in the order of 10^12^ Hz. However, the electromagnetic fields should not be treated classically, while the interaction of molecules with the ubiquitous vacuum electromagnetic fluctuations must be considered to properly understand the condensation process [37], as discussed in the following section.

### 2.2. Synthesis of the Theoretical Background in QFT-QED for Liquid Water

By adopting a QFT-QED description of liquid water (and condensed matter in general) where a complementarity relationship between the phase of oscillation and the number of oscillating quanta emerges, the HB is consistently countable for as an *emergent property* of boson condensation at a new (lower) ground level (vacuum) of water molecules. Under these circumstances, their dipolar oscillations are kept in phase by a coupled and self-trapped electromagnetic (em) vector potential (*A*), as shown in the relevant literature [37,38,39,40].

By allowing some approximations in the matter–em-field interaction to decay (such as the SVE, which is valid for a system of isolated particles along with a finite number of degrees of freedom), and by moving to the “second quantization” of QFT (which has an infinite number of degrees of freedom), new profound insights about liquid water and the nature of HB emerge. Note: According to the *slowly varying envelope* (SVE) approximation, the frequency spectrum of the “envelope amplitudes” of the em-field is concentrated on only one mode, |ω| ≪ ω***_k_*** = |***k***| (in natural units). This means that the third order time-derivative term in the equations of motion is neglected, which shows an instability of the perturbative ground state (PGS) in the matter–em-field coupling and is responsible for a departure from it towards a non-trivial solution of the equations of motion: a coherent state (see [39] for further details. 

In the QFT-QED picture, it turns out that the cohesive energy comes from the coherence energy gap (denoted as Δ_g_ or E_coh_) associated with millions of water molecules being phase-locked and packed together by a self-trapped em-field and not from the sum of individual incoherent, directional interactions as it is usually considered in the first quantization framework.

Here, we briefly summarize the two-fluid picture for water as it emerges from the QED theory, first developed in the 1980s, [37,38,39,41] by referring also to crucial experimental data that endorse this theoretical approach.

The thermodynamics of water liquefaction from vapour has been precisely treated in classical physical chemistry (allowing the quantification of the amounts of latent heat), see vol. 1 in [42], within a purely QM corpuscular picture (which, for the condensation into a liquid state, relies on the establishment of a flickering network of local, directional intermolecular forces, the so-called “HBs”). However, the real physical origin of such high values of latent heat of vaporization, boiling temperature and, above all, entropy variation cannot be fully deduced.

The main limitations of corpuscular QM regard mainly two intertwined points: (i) the inability to consider systems with large amounts of, or infinite, degrees of freedom (where the number operator $\hat{N}$, remains undefined) [43], and (ii) the inability to describe symmetry breaking (i.e., phase transitions), being a theory obeying the von Neumann’s theorem [44]. According to this fundamental theorem of QM, only one ground state, the vacuum level, is considered, rendering the description of symmetry breakings and phase transitions impossible [45]. The main problem associated with such limitations is the impossibility of predicting non-trivial solutions to the equations of motion starting from the perturbative ground state of the system (like vapour being cooled down). The vacuum fluctuations (which can excite the electron of the water molecule) are taken into account, but—because of the order of δ ≈ 1 ppm (i.e., the Lamb shift [46])—are deemed negligible. In fact, when the number of matter quanta (molecules) exceeds a critical threshold, this coupling (between vacuum virtual excitations and matter), becomes so significant that it dramatically alters the system layout simply because it is not proportional just to *N* but to *N√N* [38,39]. Moreover, in a QM picture, excitations over other levels in the spectrum of the molecule beyond the investigated transition under study are not taken into account for the evolution of the system [47]. Thus, when several billions of molecules are coupled with em-quanta of the vacuum, they are back-reformed by this new emergent state, and a purely bottom-up description (based on the mere summation of the interaction calculated over a few quanta) does not result in a truthful picture [15].

From a QFT-QED perspective, when water vapour is at the liquefaction threshold (e.g., pressure P = 1 atm and temperature T = 373.15 K), the water molecules are in constant dialogue with virtual quanta popping out of the vacuum. Based on their energy content (ΔE), these quanta can excite the electrons of the water molecule at several levels. Of course, this process does not lead to a permanent energy gain in isolated molecules since the excitation (according to the Heisenberg relation) lasts only a short time ∆τ≤h/4π∆E. The interesting aspect, when looking at the photoemission spectrum of water (see Figure 3), is that the first possible transition has an energy of about 7.5 eV, and the other energetic levels are placed at >10 eV. This means that the spatial range of such excitations (the wavelength, λ, of virtual photons) is at least in the order of 100 nm (i.e., about a thousand times larger than the water molecule itself).

The numerical density of water vapour, at its boiling point (T = 373.15 K at P = 1 atm), is 2 × 10^19^ molecules⋅cm^−3^, means that an em-excitation capable to bring a water molecule to a different electron configuration, includes within its own volume (V~λ^3^) about 20,000 molecules. The more the density is increased (e.g., by lowering T), the greater the probability that the photon released by a previously excited molecule—originally adsorbed from the vacuum—will then be re-absorbed by another one. At a critical density the photons of other molecules in the affected volume become involved in the same dynamics, until a sizeable em-field is established and ultimately self-trapped in an ensemble of water molecules that is steadily growing. In the process, it sucks in millions of molecules until the volume is filled. This saturation level is determined by short-range forces where the molecules are tightly packed (intermolecular distance > 3.1 Å, which is greater than the molecular radius > 1 Å), resulting in an increase in molecular size [38]. All photons are in phase among each other and with the molecules oscillating between the two electron levels. This new ordered state occurs because it is thermodynamically more favourable, provided that the system is open and can dissipate excess energy as heat (entropy). For liquid water, it has been calculated that the energy difference, the energy gap, Δ*_g_*, is 0.16 ± 0.05 eV [39]. This is the origin of the high latent heat of liquefaction, where an excess of energy compared to that received from the original vacuum, is handed back to the environment. In water, QED calculations, using several possible candidate levels of its spectrum (as depicted in Figure 3), showed that this probability becomes 100% for the 5d level (at 12.07 eV above the ground state), above the density threshold ρ_c_ ≥ 0.32 g/cm^3^. The selection of this level as the preferred one to settle a coherent excitation, which supports the formation of the liquid phase, also involves other parameters such as: excitation energy or frequency (*ω_q_*), coupling constant between em-field and oscillating charges (*g*), photon mass renormalization term (*µ_r_*), oscillator strength (*f_q_*), renormalized frequency (*ω_r_*), energy gap (Δ*_g_*), mixing angle (α) (see Tables 1 and 2 in ref. [39] for details).

The original frequency, *ω_q_*, of the excited em-field, which now spends part of its lifetime as excited molecules (rather than as a free field) is renormalized to a lower value, *ω_r_*, associated with the phase variation and locking between the em- and matter-field. This renormalization implies that the field consists of quasi-particles—according to the Anderson-Higgs-Kibble mechanism, they have an imaginary (negative squared) mass [48]—and thus unable to propagate outside the region where the coupling with the matter-field is in force [49]. Such a region is termed “coherence domain” (CD) and represents a self-generated sub-radiant cavity of the trapped fields [50]. It is appropriate to speak of “matter-field” since the CD is an open system where molecules are constantly crossing in and out of the CD and the number (*N*) of matter quanta is undefined, enabling the phase (φ) to be well-defined. This is consistent with the QFT complementarity relation that holds for phase and number operators, such that the greater the uncertainty (∂) in terms of number, the better defined is the phase. Thus the “fundamental uncertainty relationship” is expressed as: $\partial \varphi \cdot \partial N\geq \frac{1}{2}$ (in natural units, where ℏ = c = k_B_ = 1) [37].

The expulsion of a large amount of entropic energy per molecule (which tells us a lot about the physical origin of the latent heat of liquid water condensation) places the coherent ensemble of molecules (namely, about 6 million per CD) in a lower, more favourable, ground state (vacuum level) whose energy difference with respect to the isolated molecules’ is denoted as the energy gap, Δ*_g_*. This energy difference is a crucial quantity of the system, expressing its thermodynamical stability and defining how much energy must be spent expended to release a molecule from a CD:(5)$${∆}_{g}=\omega_{q}\left[A_{0}^{2}\left(1+2\mu_{r}\right)+{sin}^{2}\alpha -\frac{3}{2}gA_{0}sin2\alpha \right]$$
where *g* is the coupling constant between em-field and matter, *µ_r_* the photon mass renormalization term, and *A*_0_ the maximum amplitude of the em-field (i.e., the em-vector potential). The renormalized frequency, *ω_r_* is defined as a function of the energy (i.e., frequency) of the excited virtual photon (equal of the energy difference existing between sp^3^ and 5d orbitals):(6)$$\omega_{r}=\omega_{q}\left|1-\dot{\phi }\right|\Rightarrow \omega_{r}=\omega_{q}\sqrt{1-g\frac{sin2\alpha }{A_{0}}}<\omega_{q}$$
whereby the phase variation of the em-field, $\dot{\phi }$ is defined by the phase-factors time derivatives of the ground (${\dot{\theta }}_{0}$) and excited (${\dot{\theta }}_{q}$) states of the matter-field. The em-field is characterized by:(7)$$\dot{\phi }={\dot{\theta }}_{q}-{\dot{\theta }}_{0}=\frac{gA_{0}}{\tan\alpha }-gA_{0}\tan\alpha$$
where α is the mixing angle between the two levels (fundamental, “0”, and excited, “q”), such that 0 < α < π/2. These calculations are performed in natural units (where ℏ = c = k_B_ = 1, and the elementary electric charge e = 0.302814). As shown in Figure 4, the coherent state results to be a time-weighted average of the new ground state (at 90%) and the excited one (10%), where the molecules adopt an expanded shape (due to the larger 5d volume). This has two important implications: (i) the coherent fraction is less dense than the interstitial vapour-like, incoherent one, and (ii) the rearranged shape of the water molecules provides a physical explanation for the electron-cloud protrusions necessary for some tetrahedral arrangements observed in some water systems by appropriate techniques [51,52,53].

Of course, as the temperature drops further, the whole system becomes fully coherent: this only occurs at T < 220 K [54]. At higher temperatures the system is composed of two populations of molecules (those gathered within CDs, *F_coh_*(*T*) and the non-coherent ones forming the vapour-like phase interstitially between CDs, *F_inc_*(*T*)). Together the relative quantities obey a sum-rule: *F_coh_*(*T*) + *F_inc_*(*T*) = 1. The higher the temperature is, the lower the coherent fraction, so that thermal agitation can gradually erode more molecules from the periphery of CDs, reducing their diameter. At room conditions (T = 300 K), *F_coh_* ~ 40%, the effective size of the CDs is about 60 nm (Figure 5).

It is also worth noting that the coherent oscillations imply that 10% of the electrons in each CD remain very close to the ionization threshold (which is at 12.62 eV, about 0.5 eV above the 5d orbital). Thus, with about six million molecules per CD, there are about 0.6×10^6^ quasi-free electrons per CD. These electrons circulating at the periphery of a CD, are subject to a repulsive ponderomotive force (see Equation (10)) and are coherent; therefore, they cannot dissipate energy neither by thermal relaxation, nor by friction (giving rise to closed supercurrents, the so-called cold vortices). The implications of this aspect are huge, but beyond the scope of this work, for which we refer to [55,56,57,58,59].

**Figure 5 ijms-25-03846-f005:**
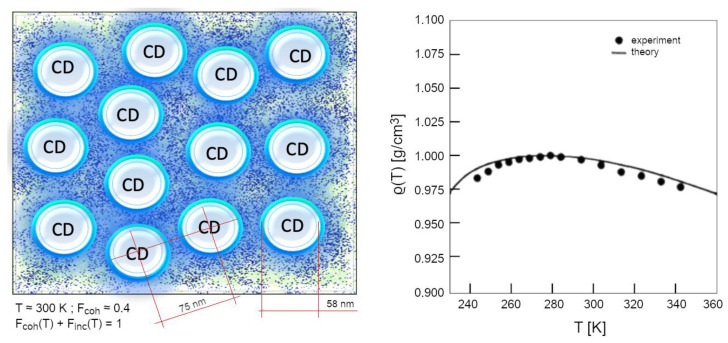
On the left, an artist’s sketch to give an idea of the biphasic picture of liquid water: at room conditions the coherent fraction, *F_coh_*(*T*), comprises about 40% of the total molecules and its density (0.92 g/cm^3^) is independent of temperature, while the density of the incoherent interstitial fraction depends on T. This requires, as shown in the right panel, modelling water density as a function of T, *ρ*(*T*). This can be done via the sum of two contributions (one for each fluid, coherent and bulk), as has been successfully demonstrated also for predicting trends of other properties in water systems (as isobar specific heat [38,54] viscosity [60] and electric susceptivity [61]).

The existence of an incoherent fraction, above 273.15 K allows water to be a liquid, whereas below that temperature the mobility is too low, causing crystallization. However, the global density, *ρ*(*T*), is lower, just because *F_coh_*(*T*) is increased with respect to 277.14 K (4 °C), at which water reveals its maximum density (Figure 5). This far outweighs the contribution of the increase in density of the incoherent fraction due to the decrease in T from 277.15 to 273.15 K. Only by applying such a two-fluid model it is possible to predict from first principles the peculiar density trend of liquid water [37,38,54]:(8)$$\rho \left(T\right)=0.92\cdot F_{coh}\left(T\right)+\left[1-F_{coh}\left(T\right)\right]\cdot \rho_{n}(T)$$

The density of the coherent fraction does not depend on the temperature and has been calculated to be 0.92 g/cm^3^ based on the broader shape given by the mixing angle weights over the coherent oscillation, *ρ_n_*(*T*) is the density of the normal (incoherent) fraction (*F_inc_*(*T*) = 1 − *F_coh_*(*T*)). See [15,38] for further details.

Recently some important experimental data have showed both (i) the impossibility to explain some features within the corpuscular-QM picture and (ii) the necessity to consider coherence as a key property within the models in order to fit the experimental data correctly.

The first case refers to infrared (IR) and near IR (NIR) analysis of water or water solutions spectra (of O-H stretching region, IR, or of its first harmonic, NIR) taken at different temperatures [62,63] whose trends showed the clear existence of an isosbestic points, expressing the existence of two populations of molecules that depend on T in a reciprocal way. This, of course, is not a novelty, but what is worth looking at is the resulting van’t Hoff plots (i.e., the Log (equilibrium constant of the transition from one population to the other) vs. 1/T) are linear, revealing that (i) the energy difference between the two states does not depend on T and (ii) that its slope is in good agreement with the energy gap predicted by QED theory. Furthermore, in [63] it has been shown how the plot of the logarithm of the ratios between the spectral intensity of one population (distinguished from the other one by the isosbestic point) with respect to the total, taken at each temperature, plotted as a function of log T yields a straight line. This accounts for a scale-free behaviour, revealing the coherent dynamics underlying the demonstrated isomorphism existing between self-similar (fractal) topologies and squeezed quantum coherent states [64,65,66].The second case deals with fitting of the dielectric permittivity of pure water and electrolyte water solutions in the range 0.2–1.5 THz [61,67]. The fit to the experimental data requires a two-fluid Debye model that mimics the electrical permittivity (both for the real and the imaginary parts). However, in order to be effective over the full spectral range, it requires an additional linear term (ξω, where ξ ≈ 0.47 ps) to the imaginary part of the dielectric function. This fact has a profound physical meaning because it implies the violation of the Kramers–Kronig (KK) relation [68] within the time span ξ. The KK relation expresses the causal relationship between the forcing field and the charge displacement. This tiny violation, on a time scale of the order of the renormalized oscillation period of the coherent field within the CDs (which excite and relax in a few hundred femtoseconds, τ_r_ ≈ 1/ω_r_ ~ 300–500 fs) witnesses temporally non-local correlations in the medium (i.e., phase correlations), possible when the system is in an entangled coherent state (a phase eigenstate). As Li has pointed out [69,70], the concept of coherence is closely related to Heisenberg’s uncertainty principle, i.e., *coherence space-time* being actually equivalent to the *uncertainty space-time*. This is the region of space and time within which particles lose their classical properties of individuality and countability (the $\hat{N}$ operator becomes undefined). The particles and fields within the coherence space-time region must be considered as an indivisible *whole* in which the phase is well-defined: thus, what happens to “a part” of a CD, within its coherent space-time range, happens to the whole CD [69]. This is a noteworthy point also for overcoming the prevailing naïve picture of the HBs [40], which are still conceived as forces acting between “particles”. As described, this classic idea originated from the first quantization can be fruitfully replaced by the QFT perspective (second quantization) where the apparent (non-directional) force is the emergent property resulting from an energy gradient that is NOT primarily bound to the binding between molecules, but is established at the ground energy level (vacuum) [37] as a consequence of the “em-field + matter-field” coupling over the whole high-numbered system [39].Another crucial issue concerns ions and their solvation in water. Within an electrostatic conception of the dissolution of electrolytes in water, the initial dynamics has no physical consistency, since few layers of water molecules should be able to keep some Na^+^ and Cl^−^ ions away from their crystal lattice, if the energy barrier to be overcome—in order to break the ionic bond—is in the order of 5 eV. A single water layer could at most produce a dielectric drop in the Coulomb force equal to 13 (ε_r_ = 13 and not ε_r_ = 80 which applies to bulk water). Again, only by abandoning an ingenuous “sticks-and-balls” interpretation of condensed matter, and by taking into account the quantum electrodynamic nature of objects like ions and their coupling with the vacuum, is it possible to consistently describe the spontaneous solvation process, showing that ions in the incoherent fraction of water establish their own coherence domains with their energy gaps (larger than the ion-bond energy) and dissolve in the liquid phase of the solvent without collisions [17]. This explains (i) why the solvent power of water increases with rising temperatures (although the net value of the bulk dielectric permittivity decreases), (ii) why there is no emission of bremsstrahlung radiation from an electrolyte solution and (iii) why the phenomenon of ion-cyclotron resonance occurs [71].There are numerous other cases, which we will only briefly mention here, as they are beyond the scope of this topic and, therefore, will be dealt with in future papers. These include the morphogenic role of water in biological matter [72], interfacial water [73], dispersion properties of biologically bound water upon exposure in the 10 Hz to 100 GHz range [74], floating water bridge [75], burning salt water upon RF-exposure [76], branching chain reaction of water [77], coherent water and cellular information processing [78] as well as stable water mixtures of both hydrophobic/hydrophilic liquids [79].

## 3. Discussion

The picture of liquid water, the paramount “HB-based” liquid in the commonly shared vision, now takes on a very different portrait, by including electrodynamic coherence, which is able to account for many non-trivial properties and their dependence on temperature.

To better understand this new paradigm (as depicted in Figure 6) as a result of the second quantization, it is necessary to focus on the nature of the cohesive forces within the coherent system and to delve into some details of the em-field dynamics within the CD.

Molecules resonating inside a CD are forced to stack on top of each other as far as their short-range repulsive forces allow, because (i) from a thermodynamic perspective, they are forced to settle in a ground state lower than that experienced outside a CD when uncoupled from the coherent em-field; and because (ii) from an electrodynamic perspective, the more quanta oscillate in phase, the better the phase is defined and, again, the more stable the coherence is. This means that the energy gap is deepened [41]. This second aspect is also responsible for the cohesion among CDs, resulting in a macroscopically condensed system, like a liquid at room conditions, rather than a gas [38,39]. The other important notion underlying CDs’ cohesion concerns the emergence of long-range dispersive forces from the self-trapped em-field gradient along the radial distance from centre-to-periphery inside a CD [39]. These forces are developed into two contributions: one frequency-independent electrostatic (Equation (9)), the other frequency-dependent electro-dynamic (Equation (10)):(9)$$F=-\frac{q^{2}}{M}\nabla A^{2}\approx qV\;(V≅0.1Volt)$$
(10)$$F_{i-CD}=C\frac{\left(\omega_{CD}^{2}-\omega_{i}^{2}\right)}{{\left(\omega_{CD}^{2}-\omega_{i}^{2}\right)}^{2}-\Gamma^{2}}\nabla A^{2}$$

The upper term (Equation (9)) expresses the ponderomotive force, which shows that any electrical charge (regardless of its sign) at the surface of a CD is repelled outwards proportionally to the ratio q^2^/m. This can be observed with the quasi-free electrons of the cold vortices confined to the outer surface. Electrons, being about 2000 times lighter than protons, are thus repelled outwards much more strongly than protons. This means that any molecule able to reach the surface of the CD is put in a polarized and, therefore, unstable state, immediately becoming a more available species for possible chemical reactions [56]. On the other hand, (Equation (10)) expresses how an *i-th* system (composed of electrical charges, but not necessarily having a net charge), capable of oscillating at a frequency *ω_i_*, very close to that of the CD (*ω_CD_ ≈ ω_i_*), is selectively attracted (if the difference *ω*^2^*_CD_* − *ω*^2^*_i_* is positive) or repelled (if the difference is negative) to the surface of the CD. In other words, if a species (such as ions or any molecule, even without a net electrical charge) has a vibrational mode whose frequency is very close to the frequency of the CD, it is selectively subjected to a diverging force (typically interesting if it is attractive) which vanishes as the square of the frequency difference decreases. Normally, the CDs experience the same boundary conditions, so they oscillate at the same frequency. This leads to cohesion of the fluid on the macroscopic scale [38,39].

Summarizing what has been said so far, it is clear that:water is necessarily a two-fluid system, like already proposed by Röntgen more than a century ago [81];the two phases in liquid water differ from one another for much deeper physical reasons than “different arrangements” (moreover unjustifiable) of the classical “HB-networks”;the short-range (electrostatic or perturbatively electrodynamic) forces—such as van der Waals interactions—act mainly in the non-coherent fraction and do not change their typicality depending on the aggregation state (clusters, normal liquid, supercooled liquid, types of ice, etc.) and, together with the long-range forces, determine the maximum achievable close-packing density in coherent fraction;the main cause of the cohesion of the system cannot be attributed primarily to local, directional, short-range forces among molecules (which, even if attractive, would not be sufficient at room temperature [14,39,40]). Instead, the emergence of a coherent matter field consisting of in-phase oscillating electric charges and photons creates potential wells (as large as the volume of the photons) at the ground level (vacuum) that are experienced by nearby molecules. An analogy can be made with marbles placed on an elastic cloth, which cluster together in the depression created by their own weight (if they are sufficiently close to one another, i.e., dense enough), and not by the existence of a net attractive force between them. Coherence causes water molecules to flip into such a minimum potential energy well, see Figure 7;the differences found experimentally in the emergent intermolecular “attraction”, called “hydrogen bonding” in a QM-corpuscular perspective, are due to the dependence on the energy-well profile within the CD. In this way, we can understand why this apparent “intermolecular” force depends on the thermodynamic boundary conditions and on the type of aggregation experienced by the molecules (see Figure 8).

Let us explain these last two points in more detail. The energy gap, namely, is independent of temperature and of external conditions since it depends only on intrinsic physical quantities of the system and on the field amplitude, which is described spatially in two regimes (inside and outside the CD, whose radius is *r_CD_*) as follows [39]:(11)$$\mathrm{For}\;\left\{\begin{array}{c} r<r_{CD}\Rightarrow {A\left(r,t\right)=A}_{0}\cdot \frac{\sin\left(\omega_{q}r\right)}{\omega_{q}r}\cdot exp(-i\omega_{r}t)\\ r>r_{CD}\Rightarrow \frac{d^{2}(rA)}{d^{2}r}-((\omega_{q}^{2}-\omega_{r}^{2})/rA)=0 \end{array}\right.\Rightarrow \;\Rightarrow \left\{\begin{array}{c} r_{CD-eff}=\frac{3}{4}\pi \frac{1}{\omega_{q}}\\ A\left(r\right)\approx \frac{A_{0}}{\sqrt{2}}\cdot \frac{exp\left[-\sqrt{\omega_{q}^{2}-\omega_{r}^{2}}(r-r_{CD-eff})\right]}{r\omega_{q}} \end{array}\right.$$

Using a dimensionless spatial parameter, 0 < *x* = *r ω_q_*/*π* < 3/4 (scaled to the CD radius, in natural units *π*/*ω_q_* = *λ_q_* ≈ 2*r_CD_*), the decay of the self-trapped em-field, from the centre to the periphery of the CD, is modulated by an envelope function *F*(*x*), expressing its exponential damping:(12)$$A\left(x\right)=A_{0}\cdot F\left(x\right)=A_{0}\cdot \left[-\frac{sin\pi x}{\pi x}+\frac{\sqrt{2}}{\pi }\cdot \frac{\exp\left[-\pi (\frac{3}{4}-x)\right]}{3-2x}\right]$$

The energy gain *E_g_*(*x*), as a function of the radial distance x, follows an envelope analogous to the self-trapped field (since the former is built on the latter, see Figure 7):(13a)$$E_{g}\left(x\right)={∆}_{g}\cdot g(x)$$
with
(13b)$$g\left(x\right)={\left\{F(x\right\}}^{2}={\left[-\frac{sin\pi x}{\pi x}+\frac{\sqrt{2}}{\pi }\cdot \frac{\exp\left[-\pi (\frac{3}{4}-x)\right]}{3-2x}\right]}^{2}$$

We can now make our final considerations. The common misunderstanding around the generally accepted view of condensed matter [37] is embedded in the limited perspective from which the problem is considered, as it neglects part the coupling terms between matter- and em-field [37,43]. To explain how the apparent intermolecular forces and bonds arise in QFT, we use a toy-system (as shown in lower part of Figure 8).

When we look at the condensed state of matter (e.g., liquid water) and try to separate one molecule from another, we expect to detect a force, a “bond”, that opposes our distancing action. The misinterpretation within the corpuscular view is rooted in the assumption that this “bond” is a real force acting ‘between’ molecules: it is assumed that molecules are the same as when they are in their isolated state, with the same structure and individual properties, and that such a “force” becomes relevant when their relative distance is sufficiently small. Actually, it is true that the funnel-like deformation of the elastic cloth (metaphor of a gradient in vacuum level) caused by the same weight of the sufficiently many marbles (Figure 8) will force them to cluster together at the bottom. The water molecules inside the CD just tend to occupy the same minimal potential state (i.e., a lower ground state achieved thanks to the coupling between matter and em-field (unpredictable in a merely perturbative regime as the corpuscular view of the QM pair-potential [14]). Thus, the apparent “force” we would experience when trying to separate some marbles from some others, is the result of the potential gradient, their thermodynamic tendency, to remain in the minimum available energy state, as outlined by the deformed cloth (which is the alter ego of the vacuum, the ground state). This gradient, represented by an “invisible” elastic cloth, is transduced into a “force”, into a ‘bond strength”. Yet, the marbles do not attract to one another if placed on a rigid floor (except for perturbative regimes, which are ridiculously small and, therefore, negligible as for molecules this force is in the order of 1/√N [39]).

The elastic cloth is the “allegory” of the vacuum, the ground state; while the fact that it can be deformed (and change its level) is a metaphor for the violation of von Neuman’s theorem, i.e., many vacuum levels are possible within the framework of the second quantization (QFT), but not within the first quantization (QM). The fact that the marbles, when numerous enough, cause the elastic cloth to bend is a figurative representation of the coupling between matter and em-field, which becomes relevant once a density threshold is reached. The resulting deformation, just like a black hole, attracts more marbles until the pit is densely populated, corresponding to the sub-radiant lasing cavity called coherence domain. In the case of water, such self-sustaining dynamics is produced by the numerical density of molecules which, once it exceeds a critical threshold, can only attract more molecules (up to a saturation level given by short range forces and by the volume size of the CD, which is governed by the wavelength of the self-trapped em-mode) [38]. A characteristic feature of this ensemble is that they start to oscillate all in phase (tuned by the now trapped em-field) shifting into the so-called runaway escalation leading to liquid condensation [38]).

Of course, in phenomenological terms, we could say that a “binding force” between molecules appears and that this apparent force varies with the system boundary conditions (i.e., macroscopic parameters like pressure and temperature). However, having this electrodynamic picture at hand, we are able to understand the physical meaning of this and we can consistently explain the fact that in ice the resulting calculated “binding force” is dramatically larger than in a cluster of supercooled confined liquid water or in a putative “water dimer” [19,39,40].

The degree of coherence, and the establishment of other possible coherences (associated with other available levels in the water electronic spectrum, see Table 3 in [39]), determines the overall strength of the cohesion (manifested as “bond strength”). The degree of coherence (for each) is expressed by the electrodynamic functions of the field amplitude and the energy profile as a function of radial distance from the centre of the CD, and these functions also depend on temperature [54]. And on the macroscopic level their percentage is expressed by the sum rule: *F_coh_*(*T*) + *F_inc_*(*T*) = 1. From this it can be qualitatively deduced that the strength of any kind of coherence experienced by matter quanta (molecules) is not uniform over the whole CD volume, but rather depends on their radial position within the CD and is thus influenced by the shape of the ground state energy gradient (see, Figure 9). The resulting overall “bond strength” is thus the resultant of all contributions acting on the system, such as the possibility of coherence also occurring for other modes, as outlined for ice and low temperature states of water in [39,60] (and possibly not even regarding electrons, but, for instance, nuclei positions, like for the crystalline states [37]). This explains why the estimated “HB-strength” is higher in hexagonal ice of the type I_h_ than in the amorphous state [82].

With this contemporary interpretation at hand, it is also easier to explain the 75 known water anomalies [83], which would still remain a mystery when addressed within first quantization. Within second quantization, however, these various peculiar properties of water will suddenly lose their mysterious veil, once each of these anomalies undergoes vigorous testing within the QFT-QED framework.

## 4. Conclusions

In this review, we wanted to address the problematic HB concept that is still based on the corpuscular “sticks-and-balls” view describing numerous condensed matter systems (from liquid water to biological molecules). We showed that within the general QM-theoretical approach (first quantization), it is not possible to derive a consistent physically based description of the underlying dynamics that such a supposed “bond” should have. The usual picture of condensed matter to interpret the experimental data obtained in physical chemistry and spectroscopy, for example, continues to emphasize this paradigm. The result of such a view is a conceptual dichotomy: the QM-corpuscular description of matter (where the matter–em-field interaction is described by considering the interaction with vacuum just as a perturbative order), provides, on the one hand, a picture in which molecules (like H_2_O) exist in physical states (electron cloud shapes) that are unable to satisfy the same conditions that would be necessary to produce those cohesive forces (like HBs). On the other hand, the HB can no longer be justified in its peculiar dependence on the thermodynamic boundary conditions of the systems, since, within the same purely QM picture, it would always remain the same (like van der Waals forces).

We have focused our attention on liquid water, but the general ideas can be applied to all condensed systems [37,45,50]. Having addressed the main problems posed by the QM-corpuscular view, we have reformulated key aspects of the HB-concept by applying CD-structures as they emerge from the QFT-QED theoretical description developed since 1988. In the case of liquid water, it is sufficient to reframe the challenges of condensed phases of matter together with the “HB-concept”, as well as spectroscopic data within a “field view”. The two-fluid model of water within second quantization explains not only the boiling point shift from −150 to +100 °C but also the density anomaly observed at +4 °C. It also accounts for the strong cohesion of crystalline hexagonal (I_h_) ice, where coherence is established over molecular positions [37], and the supercooled liquid state [72]. In fact, using the QED approach, quite a number of anomalous behaviours of water no longer need to be classified “anomalous” because they appear as a logical consequence of the principles applied within the QED description. Finally, having clarified that liquid water is not just a two-fluid system—as postulated by [84]—but one with a coherent and an incoherent phase, we have outlined a QED interpretation for HBs, supporting it with calculations, and graphical intercomparisons, in order to allow this powerful, eye-opening view and its potential consequences to be seized and discussed by as many scientists as possible.

## Figures and Tables

**Figure 1 ijms-25-03846-f001:**
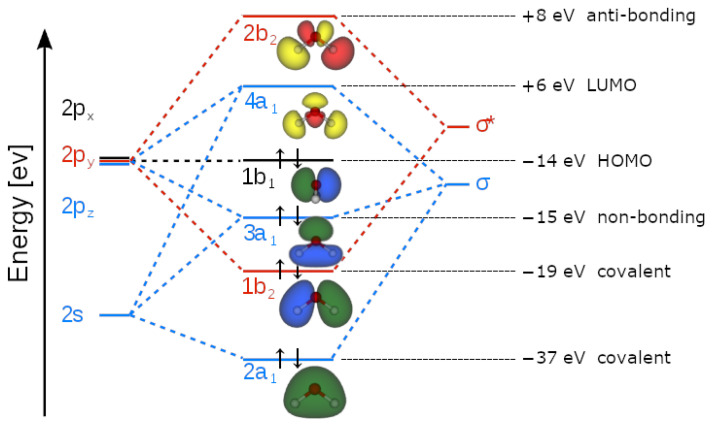
Molecular orbitals of the water molecule in LCAO theory of the isolated molecule. Note: the inner 1s orbital is not shown; σ denotes the bonding configuration; σ* denotes the anti-bonding configuration, which leads to molecular instability and thus splitting of the constituting atoms. The colours represent the positive (green/yellow) or negative (blue/red) value of the orbital wave function participating to the exchange integral describing the bonding character (positive product) or antibonding character (negative product) of the interaction (product) with other orbitals associated with the estimated locations of the electrons around the nuclei (Composite representation based on [24,25]).

**Figure 2 ijms-25-03846-f002:**
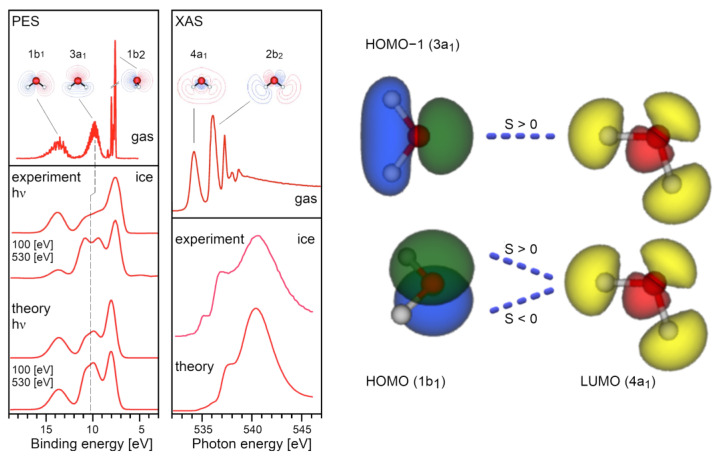
(**Left**) Panels: Gas phase Photo-Electron Spectroscopy (PES) spectrum of water measured at a photon energy of 100 eV (top left) and PES spectra of ice at photon energies of 100 eV and 530 eV (bottom left). X-ray Absorption Spectroscopy (XAS) spectra of gas phase water and ice together with a Density Functional Theory (DFT) calculation of hexagonal ice described by a 44-molecule cluster [26]. (**Right**) Sketch: Scheme showing that the 3a_1_-(HOMO-1) molecular orbital lies in the molecular plane, while the 1b_1_-(HOMO) orbital lies in a plane perpendicular to the σ-bonds (O–H). Yellow-blue pairing denotes anti-bonding, yellow-green pairing denotes bonding orbitals. The overlap is expressed by the exchange-integral S. Partial HB’s covalence is thus only possible if the fully occupied in-plane 3a_1_-(HOMO-1) level overlaps with an empty 4a_1_-(LUMO) of a neighbouring molecule. The overlap with the other out-of-plane 1b_1_-(HOMO) is zero. Strong HBs are then expected to interact with the 3a_1_-level while van der Waals interactions are expected for the 1b_1_-level. However, the changes in the 3a_1_ orbital revealed by XES [26] are not experimental evidence for electron sharing (covalence) in HBs [29] because in (HOMO-LUMO) frontier-orbital theory the assumed covalence would primarily affect the HOMO outmost 1b_1_ orbital and definitively not the 3a_1_-(HOMO-1). Covalence in a QM sense, i.e., HOMO/LUMO interaction, is thus not supported by experimental data. In other words, a valid CP or QM picture of HB seems impossible.

**Figure 3 ijms-25-03846-f003:**
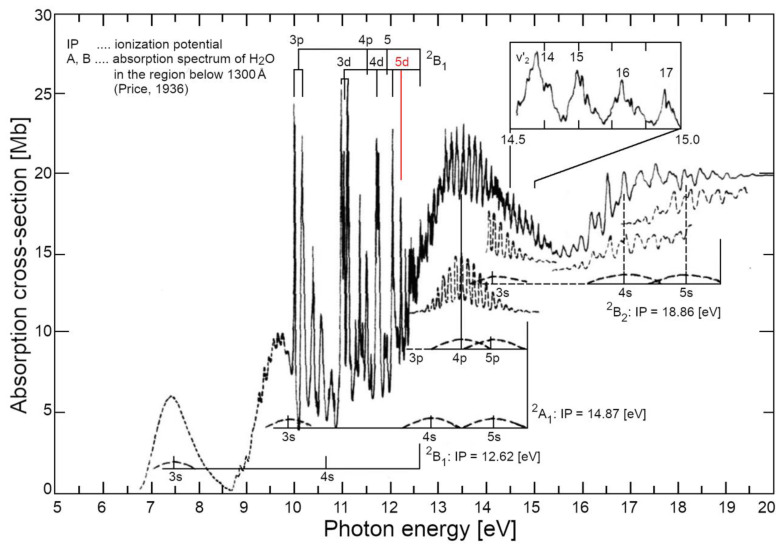
The excitation spectrum [32,33] of water vapour (isolated molecules) in the visible and near UV range (up to 20 eV); the red line indicates (by considering the final density of the liquid at room P and T) the predicted excited level (5d orbital) that satisfies several favourable conditions at once, such as not too high critical density, sufficient oscillator strength, relevant coupling constants *g_c_*, *μ_r_* and energy gap.

**Figure 4 ijms-25-03846-f004:**
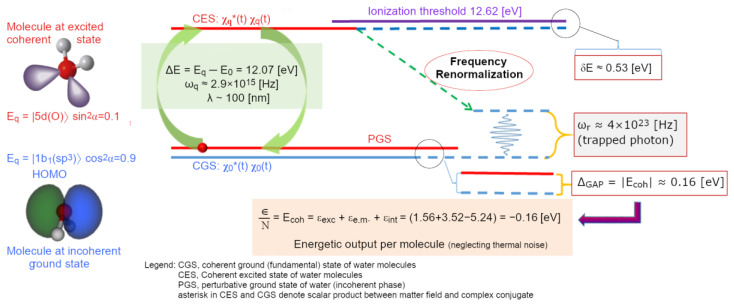
A scheme of the energetics of the coherent state of water molecules showing that such a state is the result of a continuous *collective* oscillation of the molecules between two states, driven by the self-trapped em-field whose phase (and renormalized frequency) is locked to the phase of the matter-field. Considering the coherent oscillation, the water molecules adopt two limit-shapes (excited and relaxed) with different time-weights during the oscillation cycle. The value of the energy gap, (Δ*g* ≈ 0.16 eV) in this scheme refers to the latest calculation done for liquid water, neglecting the temperature contribution [39]. However, as will be discussed in the following, the energy stability is not independent of the radial position within the CD (see Equation (13a,b) in the text), thus the energetic profile of the coherent ground state within CDs depends on the aggregation state and the thermodynamic conditions. The thermodynamic parameters (like pressure and temperature) can influence the establishment of other types of coherence [39], favouring the choice of other excited levels among those available in the electron spectrum of water (to which other field amplitude, oscillator strength, energy gap, coupling constant, critical density, renormalization frequency are associated). This is a key aspect to understand why the classical concept of “HB” depends on the aggregation state (water dimer, liquid, ice, supercooled clusters, etc.) [15]. A mixing angle α, which gives sin^2^(α) = 0.1 indicates that electrons in water molecules spend 10% of their time in the excited level (Eq, the 5d oxygen orbital), so that coherent water molecules are larger than incoherent water molecules. Such a fact can explain (i) the flickering landscape of intermolecular interactions (including the so-called “HBs”), as well as (ii) the evidence of tetrahedral structures in some regions of the liquid (or as in hexagonal ice and in confined water). Indeed, two of the five d-orbitals (z^2^, x^2^ − y^2^) transform into the totally symmetric a1-representation of the C_2v_ group and can mix themselves with the two other molecular orbitals (2a_1_, 3a_1_) giving rise to a set of four a_1_-type levels arranged in a more or less tetrahedral configuration to minimize electronic repulsions [40].

**Figure 6 ijms-25-03846-f006:**
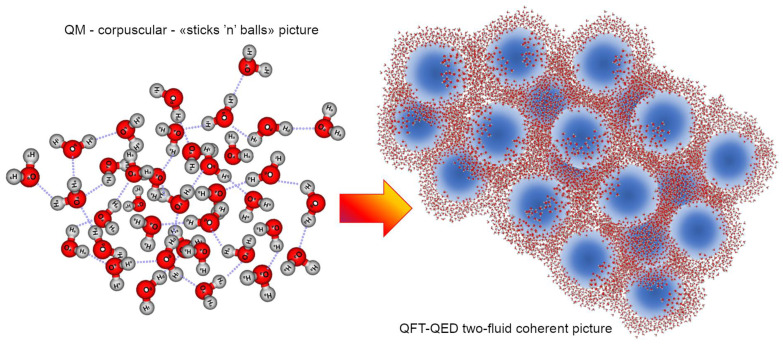
Artistic sketch to give an intuitive idea of the two different pictures emerging from corpuscular QM (**left**) and QFT-QED (**right**) views: within the former approach—where electrodynamic interactions are only perturbative—the cohesion and condensed state of liquid water is supposed to rest on a flickering network of local directional forces. This picture suffers from the inability to physically justify how water molecules could express a topology of their electron cloud that is protruding enough to yield directional electrostatic oriented pair-potentials. It also lacks an explanation of the physical reason why this flickering network in such a “mixture model” [80] should be divided into two populations (as required to fit the now common experimental evidence that liquid water is a two-phase system) [26]. On the right pane, an “instantaneous frame” representing liquid water at ordinary temperatures, where CDs (in blue), appearing and disappearing every few hundred of femtoseconds, are immersed in a stochastic, vapor-like, incoherent (denser) fraction, that becomes more and more abundant with increasing temperature. In the latter case, two types of dynamics co-exist, time-relaxations, kinetics, orderings, and geometries [40].

**Figure 7 ijms-25-03846-f007:**
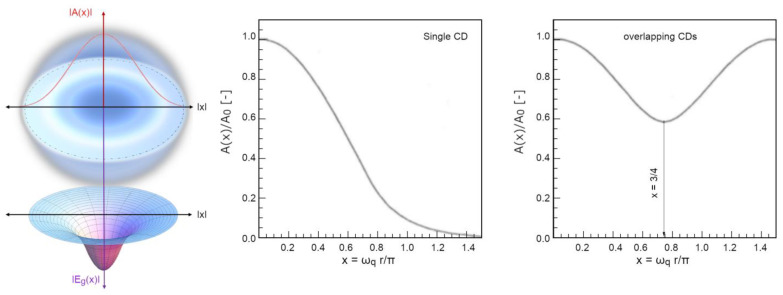
On the left panel, an intuitive sketch to describe the CD as a region of space where, the confined decaying em-field (top scheme) couples with matter quanta, generating a new, energetically lower, vacuum level (where phase correlations operate thanks to the condensation of bosonic quasi-particles). In SSB, the massless part of these quasi-particles is called phasons [50] and to their space range a lower vacuum level is associated, constituting a potential well with respect to non-coherent region. The passage from a vacuum level to the other traces an energetic profile across the CD boundary (lower scheme). This potential well is populated by coherent molecules that experience a lower energy in the fundamental state than in the isolated, vapour state. The right panel depicts the profiles of the reduced field amplitude, *A*(*x*)/*A*_0_, as a function of the radial parameter x, within the CD (reported from [38]). The centremost panel illustrates a single-CD decay profile, whereas the right panel shows the overlapping field amplitude between two adjacent CDs.

**Figure 8 ijms-25-03846-f008:**
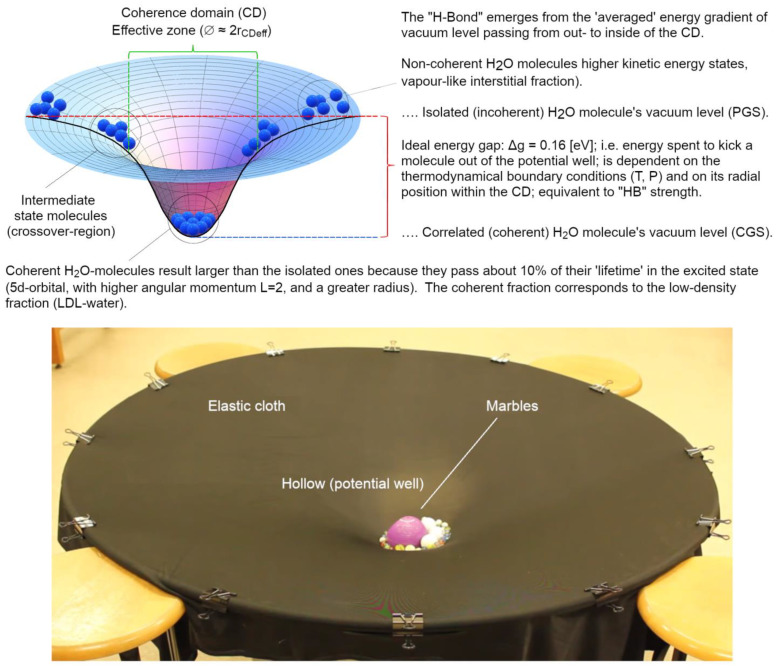
The top panel represents the different ground states (vacua) experienced by water molecules (schematized as blue spheres) inside and outside the CD. Due to the em-field confinement, a potential gradient is established across the CD interface: the scheme serves to illustrate how the effective thermodynamic stability enjoyed by molecules belonging to the CD also depends on the radial position within the CD itself. Thus, the thermodynamic stability of coherence, which is usually derived from experimental data as different kinds/strengths/arrangements of “HB”, is the averaged result of many potential depths experienced by molecules depending on their radial position and on the width of the CD. What is considered to be a force existing between molecules (as generally conceived within the QM view of the first quantization), and which becomes macroscopically manifest as condensation when a critical density is reached, is now understood metaphorically to be exactly the same “apparent force” that we face if we tried to separate some marbles from each other resting at the bottom of the depression of an elastic cloth. This “force” is not something that exists intrinsically among the marbles. It is an emergent property that manifests itself through the coupling between the marbles and their (new) vacuum (ground) level, represented by the deformation of the cloth (second quantization). See the text for more details.

**Figure 9 ijms-25-03846-f009:**
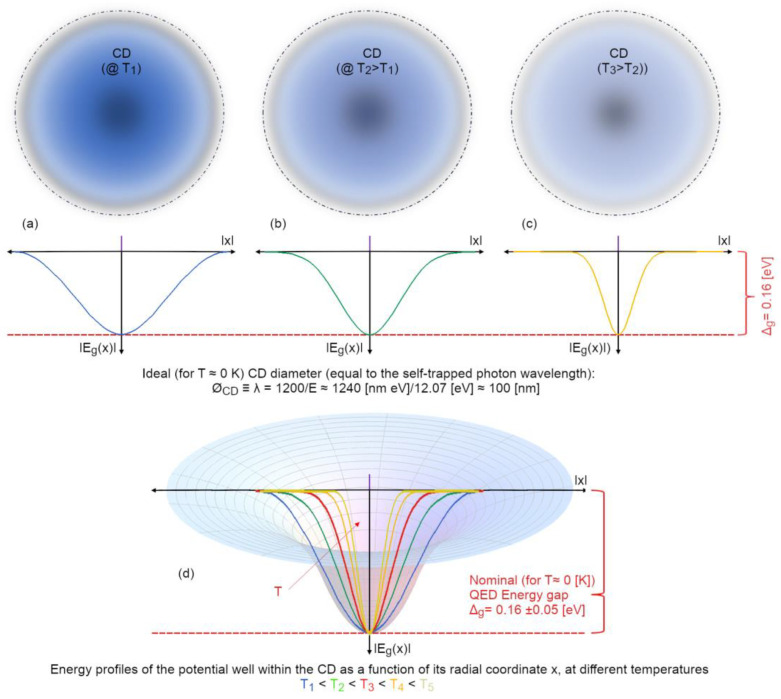
Scheme of the qualitative change in the profile gradient at ground level for three different temperature values T_1_ < T_2_ < T_3_ (grouped together in the upper panels (**a**–**c**)). The maximum depth of the well, predicted within the framework of theoretical QED, is independent of temperature, Δ*_g_* ≈ 0.16 eV. The higher the thermal erosion and the narrower the energy cone-shaped well of the CD becomes (thermodynamically less stabilized), the more molecules (primarily at CD boundaries) experience weaker coherence. Increasing the temperature is comparable to reduce the available peripheral volume of the CD (making it narrower) while coherence is still strong (good potential depth). Panel (**d**) shows a sketch for 5 temperatures). A peculiar feature is that the profile of the energy-well becomes steeper and steeper as T increases, making it more difficult for molecules to “jump” out of the narrowing pit: such dynamics prevent an avalanche process that might otherwise lead to easy destruction of the coherent phase during evaporation. In fact, many CDs persist, in a reduced size, even in the gas phase, and to dissolve them completely, temperatures well above the thermodynamic boiling point of around 600 K must be reached [54].

## Data Availability

Not applicable.

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
