# Peer review of "What Is the “Hydrogen Bond”? A QFT-QED Perspective"

_ijms, 2024, doi:10.3390/ijms25073846_

Round 1

Reviewer 1 Report

Comments and Suggestions for Authors

The manuscript titled "What is the 'Hydrogen Bond'? A QFT-QED perspective" by Paolo Renati and Pierre Madl provides an innovative perspective on the nature of the hydrogen bond (HB) through the lens of Quantum Field Theory (QFT) and Quantum Electro-Dynamics (QED). It challenges the traditional view of HB as merely a dipolar force between molecules, proposing instead that it arises from electromagnetic field gradients within the coherent phase of water. This theory suggests a more complex interaction mechanism, contributing to the understanding of water's anomalous behaviors and the energy associated with HB under different molecular conditions. The manuscript elaborates on the theoretical background, experimental data interpretations, and implications for condensed matter and living systems, advocating for a shift from classical Quantum Mechanical views to a QFT framework for a more accurate description of water and HB dynamics.

However, in reviewing the manuscript, I've noted a critical area for improvement that would significantly enhance its scholarly rigor and reader comprehension. It is imperative that each reference within the manuscript be explicitly connected to the specific claims, data points, and theoretical concepts it is intended to support. This meticulous alignment is not only fundamental for validating the arguments presented but also crucial for enabling readers to trace the evolution of the manuscript's conclusions within the context of the existing literature. Ensuring that references directly correlate with the relevant segments of the text will bolster the manuscript's academic integrity and facilitate a more profound understanding of its contributions to the field.

Author Response

Rev.1: However, in reviewing the manuscript, I've noted a critical area for improvement that would significantly enhance its scholarly rigor and reader comprehension. It is imperative that each reference within the manuscript be explicitly connected to the specific claims, data points, and theoretical concepts it is intended to support. This meticulous alignment is not only fundamental for validating the arguments presented but also crucial for enabling readers to trace the evolution of the manuscript's conclusions within the context of the existing literature. 

Authors: We have checked the originally uploaded version and can confirm that the references were correct. We simply do not know how such a discrepancy could have occurred. In any case, please accept our apologies. Thus, we have gone through the references, corrected double entries (Wernet et al) and also made improvements in the main text of the manuscript.  Figure 5 has also been updated to make the distinction between experiment and theory clearer.

Reviewer 2 Report

Comments and Suggestions for Authors

This is a beautifully written, highly educative paper on the nature of hydrogen bonding. I strongly recommend its publication as it is. However, some references seem scrambled and some formatting issues have been detected. I recommend the authors double-check that referencing is ok before approving their submissions.

In this review, the author describes the concepts of hydrogen bonding well beyond the typical chemist's view of dipole interaction. They point out the complex quantum-mechanical nature of this highly important kind of bond and provide a clear perspective of the current situation. The two-fluid description of liquid water is clearly shown and, although the physical background is rather complex, the author presents an intuitive picture of the problem.

This is a highly educational manuscript. I enjoy reading it and, despite the complex nature of the problem, I believe that they give a clear view of the situation appealing to a broad audience. Is my opinion that the paper can be published as it is.

There is only a small problem the author needs to correct, which is related to formatting, I believe. In some parts of the text, references are missing, there are spurious blanks (see, for example, Page 7, second line of the third paragraph) that make it difficult to check the appropriateness of the references. However, I assume that this is a problem that can be easily corrected by the author since the list of references seems appropriate, given the contents of the paper.

In brief, after checking formatting issues, I believe that this is a notable review and it can be published in its present form.

Author Response

Rev2:  .... some references seem scrambled and some formatting issues have been detected .... a small problem the author needs to correct, which is related to formatting, I believe. In some parts of the text, references are missing, there are spurious blanks (see, for example, Page 7, second line of the third paragraph) that make it difficult to check the appropriateness of the references.

Authors: We apologise for the inconvenience caused by the misplaced references and the formatting of the core text. Thank you for pointing out a duplicate entry in the references. We have corrected the references accordingly and also made improvements in the main text of the manuscript, together with Figure 5, which has also been updated to make the distinction between experiment and theory clearer.

Reviewer 3 Report

Comments and Suggestions for Authors

Paolo Renati and Pierre Madl have presented an interesting review entitled “What is the “Hydrogen Bond”? A QFT-QED perspective.” This review covers contemporary problems in understanding the nature and role of hydrogen bonds of water, responsible for holding together its molecules in both liquid and condensed phases. The authors proclaim that the description of matter through a corpuscular approach (both classical and Quantum Mechanical, QM) does not explain unusual features of condensed systems like liquid water, resulting in abnormally high boiling point of liquid water, or the hydrogen bond energy of about 0.2 eV. In this way, the authors make emphasis on the two-fluid picture for water, as it emerges from the QED theory, and correlate the theoretical results within this model with available experimental data.

         The manuscript presents an interesting topic about the nature of water, which is on the cutting edge of scientific discussion and is still open for new ideas. The presented text is well-structured, but, apparently, hastily written: it bears a great number of misprints and errors in the references. For example, there are sentences like this: “Similarly, the seminal paper by promoted that the liquid nature …” on page 2 (paper by who?), or “… as it has been shown by relevant liter-ature ,,,” on page 7 (what literature ?). There are also a number of figures without the references to the corresponding literate sources. Such issues are abundant throughout the whole text, so I recommend the authors to perform careful editing of the text.  

    The introduction properly poses the problem, however, there are no mentioning about previously published reviews on the considered topic. Is this the first review discussing the nature of water from QFT-QED perspective?

    I suggest to extend the consideration to gaseous phase, as, even for gaseous phase, a chain of gaseous water molecules can switch from having random orientations to having highly correlated ones, and this can also be explained through quantum nuclear effects on hydrogen bonds.

       It would be interesting to discuss the quantum tunneling effect in water, which smears out the positions of the hydrogen atoms so that water molecules are turned out to be delocalized around a ring and adopt an unusual shape.

      What are the roles of anharmonic quantum fluctuations of intramolecular covalent bond stretching and intermolecular hydrogen bond bending? Is there a need for anharmonic intermolecular potentials in the force-field-based studies of quantum nuclear effects?

    In my opinion, the manuscript is worth publishing in IJMS after an extensive editing of the text and considering the issues raised by me and the other reviewers.

Comments on the Quality of English Language

The presented text is overwhelmed with grammatically incorrect phrases. There are a lot of misprints and reference errors.

Author Response

Rev3: The presented .... apparently, hastily written: it bears a great number of misprints and errors in the references. For example, there are sentences like this: “Similarly, the seminal paper by promoted that the liquid nature …” on page 2 (paper by who?), or “… as it has been shown by relevant literature…,” on page 7 (what literature ?). There are also a number of figures without the references to the corresponding literate sources. Such issues are abundant throughout the whole text, so I recommend the authors to perform careful editing of the text.”

Authors: Thank you for pointing out these shortcomings - for which we would like to apologise. We have therefore checked the final version of our submitted manuscript and we believe that during the process of delivering our document to the reviewers, some formatting problems occurred, so that the text lost some "pieces" in sentences and references. We have revised the entire manuscript and corrected any inconsistencies.

Rev3: “The introduction properly poses the problem .... there are no mentioning about previously published reviews on the considered topic. Is this the first review discussing the nature of water from QFT-QED perspective?”

Authors: We have included a short section in the Introduction to emphasise this point - see the very end of the last paragraph, highlighted in yellow.

Rev3: I suggest to extend the consideration to gaseous phase, as, even for gaseous phase, a chain of gaseous water molecules can switch from having random orientations to having highly correlated ones, and this can also be explained through quantum nuclear effects on hydrogen bonds .... and .... “It would be interesting to discuss the quantum tunnelling effect in water, which smears out the positions of the hydrogen atoms so that water molecules are turned out to be delocalized around a ring and adopt an unusual shape”.

Authors: Yes, this is a topic that deserves more attention, but we think that it would go far beyond the scope of the present work for two reasons:

1) The dipolar interactions in the gas phase remain at the perturbative level and therefore they cannot really be considered as “H-bonds” (HB), but rather as van der Waals (vdW) forces, which we have described in detail in the present work.  We hope that this will help the reader to better understand the differences between the two regimes of interaction (vdW vs. HB).  The so-called HB, considered as the interaction responsible for the cohesion of the molecules can only be attributed to non-perturbative and non-trivial states of the em-field bosons coupled to the (oscillating) electric charges of the molecules.  The HB depends on the collectivity of the correlation (hence, it varies as a function of the macroscopic thermodynamic parameters, pressure, temperature or on the phase liquid, crystalline ice, amorphous ice, etc.). VdW forces on the other hand, are electrodynamic interactions whose characteristics are given by the intrinsic parameters of the water dipoles considered for the isolated water molecule (length, stiffness, distances between molecules) and do not alter as a function of the state of aggregation.  Obviously, these interactions are not responsible for the cohesion of the liquid or ice. Nevertheless, the VdW forces could be added to the calculation of the total potential experienced by molecules when they are close enough and correctly aligned.

2) The coherence we’re studying in the present work is the one established via the oscillation of the AMPLITUDE of the water dipole (or, of its electron cloud) and NOT via the oscillation of its orientation in space (possibly with respect to a net polarization vector, as it occurs near hydrophilic surfaces, as well explained in Del Giudice et al. (2013).[1] This type of coherence is the one responsible for the emergence of the liquid (or amorphous) phase and does not concern a specific spatial order, but rather a motional order related to the phase locking of the individual electron clouds.

Thus, we are not concerned here with the spatial configurations of the water molecules, which are related to other levels of coherence (possibly coinciding with the calculated total "HB" values in condensed solid phases, such as certain types of ice - see end of section 3 of this manuscript). 

We agree that a study from a QFT-QED perspective also of the orientation of the nuclear spins and of the ortho-para configurations of the water molecules, could be very interesting, but should be treated in a separate paper.[2] Certainly, it is possible to estimate these (weaker) degrees of freedom on which other possible coherence levels can be established, as these could contribute to the overall physico-chemical properties (especially in living matter and in confined states where water molecules are already more stabilized about their translational motion).

[1] Del Giudice E, Tedeschi A, Vitiello G, Voeikov V (2013) Coherent structures in liquid water close to hydrophilic surfaces. Journal of Physics: Conference Series, 442:012028. doi: 10.1088/1742-6596/442/1/012028

[2] It is known that the distinction between "ortho" and "para" water is only possible for water molecules separated in the gaseous state. We think a good reference to start with could be: Petoukhov S (1999) Biosolitons, mysteries, living things, basic-soliton-biology. Moscow (RUS). ISBN 5-9001204-01-8, and Henry M (2016) L'Eau et la physique quantique (Water and Quantum Physics – progressing towards a medical revolution, in French), Dangles Editions. Escalquens, ISBN‎ 978-2703311478  - both extensively deal with the ortho-para-water issue.

Rev3: What are the roles of anharmonic quantum fluctuations of intramolecular covalent bond stretching and intermolecular hydrogen bond bending? 

Authors: With regard to the non-linear (anharmonic) regime of the oscillation of the of the O-H bond within the water molecule (stretching), when it experiences the collective-coherent regime, we think that it is already considered in the non-perturbative and not only electrostatic treatment of the oscillations of the electronic cloud of the molecule, which in fact also result in oscillations of the length of the covalent O-H bond.

The same can be said for the bending of the O-H covalent bond since the participation in the coherent state implies that the water molecule experiences 10% of the time a dilated shape, with the electron positioned in the 5d orbital, and a shape of the electronic cloud that is no longer rounded (potato-like), but extended (pronounced tetrahedral protrusions), as has been well expressed in Del Giudice et al., 1995. [3]

This concerns the intramolecular covalent O-H bond and means that it is in fact subject to continuous fluctuations (fractions of a picosecond). This fluctuation translates into a perturbation that affects the directionality and bending of the so-called intermolecular "HBs." However, it is important to understand that, as expressed in the previous point, within the second quantization (QFT), the liquid phase 

  1. refers to a coherence that does not concern the orientation of the molecules, but only their dipole amplitude oscillations and
  2. is considered as a matter field where the phase, but not the number, is well defined.

Therefore, we do not address the problem by talking about "one, two, three or more molecules", since such a category of countability loses its meaning in the context of coherence, simply because a coherent state is regarded as a supramolecular entity acting as a whole. This type of approach, still corpuscular, could apply (but in the perturbative regime) only to the vapour/gas phase. Focusing on the number operator, defining it, maximises the phase uncertainty and breaks the ordered relationships between molecules, leading to decoherence. Please note what we have added and highlighted in yellow in the caption of Fig. 8 of the present manuscript: "What is considered to be a force existing between molecules (as generally conceived within the QM view of the 1st quantization), and which becomes macroscopically manifest as condensation when a critical density is reached, is now understood metaphorically to be exactly the same "apparent force" that we face when we try to separate some marbles resting at the bottom of the depression of an elastic cloth. This "force" is not something that exists intrinsically between the marbles. It's an emergent property that manifests itself through the coupling between the marbles and their (new) vacuum (ground) level, represented by the deformation of the cloth (2nd quantization)". Therefore, the particle perspective is only valid in corpuscular quantum mechanics, not in QED, and can only be applied (in the perturbative regime) such as the vapour/gas phase.

[3] Del Giudice E, Galimberti U, Gamberale L, Preparata G (1995) Electrodynamical coherence in water: a possible origin of the tetrahedral coordination. Modern Physics Letters B, 9(15):953–961. doi: 10.1142/s0217984995000917

Rev3: Is there a need for anharmonic intermolecular potentials in the force-field-based studies of quantum nuclear effects?”

Authors: In analogy to the above:

  1. on the one hand, we think that nonlinear anharmonic regimes are already taken into account in the description, which departs from the perturbative and electrostatic regime. It is able to provide, from first principles, a picture that is in very good agreement with experimental evidence (such as the density trend vs temperature or the magnitude of the latent heat of liquefaction);
  2. on the other hand, we believe that nuclear effects represent an additional layer to be considered (for the liquid), in addition to the coherent regime that we have analysed and verified in this manuscript. In fact, if we think of coherence and if we place ourselves in the atomic approximation, we can see two types of discernible entities: the electrons and the nuclei. We should therefore expect three states of matter. A first state (i) where incoherence is complete at both the nuclear and electronic levels - corresponding to a gaseous state; a second state (ii) where the electrons are coherent but not the nuclei - corresponding to the liquid state; and a final third state (iii) where nuclei and electrons are in coherence, defining the solid state. For the time being, this is left to a further and more specific investigation, which will deal with this very question.

Rev3: The presented text is overwhelmed with grammatically incorrect phrases. There are a lot of misprints and reference errors.

Authors: We have gone through the entire manuscript and have indeed identified numerous inconsistencies (both grammatical and typographical) which we have corrected.  We also found an error in the bibliography, which we have also corrected.

Finally, we would like to thank the reviewer for his constructive input, which we believe has helped to improve the quality of the manuscript. Thank you very much!

Round 2

Reviewer 1 Report

Comments and Suggestions for Authors

Upon reviewing the resubmitted manuscript titled "What is the 'Hydrogen Bond'? A QFT-QED perspective" by Paolo Renati and Pierre Madl, I am pleased to note that the authors have effectively addressed the concerns highlighted in my previous review. Specifically, the critical area for improvement, which involved explicitly linking references to the claims, data points, and theoretical concepts discussed, has been satisfactorily resolved. Furthermore, the issue with the citation numbering in the initial submission, where cited content did not follow the reference publications' number, has been corrected in this revised submission. These enhancements significantly contribute to the manuscript's scholarly rigor and improve reader comprehension by enabling a clear traceability of the arguments presented within the context of existing literature. The manuscript now stands as a compelling contribution to the field, offering a novel perspective on the nature of hydrogen bonds through the lens of Quantum Field Theory and Quantum Electro-Dynamics. I commend the authors for their thorough revision and believe the manuscript is now ready for publication.

Reviewer 3 Report

Comments and Suggestions for Authors

In my opinion, all issues were addressed accordingly. The manuscript is ready for publication. Good luck in further research!